# Using photovoice to facilitate the report of emotions in an interview setting: An experimental study

Selina Studer[1], Maria Kleinstäuber[2], Cornelia Weise[1,3]*

1 Department of Psychology, Division of Clinical Psychology and Psychotherapy, Philipps-University Marburg, Marburg, Germany, 2 Department of Psychology, Emma Eccles Jones College of Education and Human Services, Utah State University, Logan, Utah, United States of America, 3 Department of Psychology, Clinical Psychology and Behavioral Health Technology, Friedrich-Alexander-Universität Erlangen-Nürnberg, Erlangen, Germany

* cornelia.weise@fau.de

## Abstract

### Background

Finding words to describe emotional experiences can be challenging. Photovoice (PV) represents a possible way of facilitating the report of emotions. In the PV approach, people take pictures that they feel are related to a certain topic. After-wards, they are invited to talk about this topic based on the pictures. There is a lack of experimental studies investigating the effectiveness of PV to aid in emotional processing in comparison to other methods.

### Methods

Sixty-five participants were randomly assigned to one of three groups (mean age: 28.23 years [$SD = 9.23$], 76% female). The photovoice-group received the task to take pictures that reflect future worries about something in the future. Participants in the active control group, the writing group, were instructed to write down their future worries. The control group did not receive an assignment. In a subsequent semi-structured interview, all participants were invited to talk about their future worries. The interviews were audio-recorded and later transcribed. The number of emotion words was counted during the transcriptions. After the interview, participants completed an online self-report questionnaire addressing a range of variables such as the partici-pants' emotional state and their perceived difficulty identifying emotions.

### Results

Contrary to our hypotheses, one-way ANOVAs revealed no differences in any of the self-report measures between the three study groups (all $p$ values $> 0.14$). Planned contrasts regarding the transcribed interviews revealed, however, that the PV-group

**Data availability statement:** All relevant data for this study are publicly available from the OSF repository (https://osf.io/2bj63/).

**Funding:** The author(s) received no specific funding for this work.

**Competing interests:** The authors have declared that no competing interests exist.

reported more negative emotion words compared to the two other groups $t(62) = 2.79$, $p = .007$, and also compared to the WG only, $t(62) = 2.48$, $p = .016$.

## Conclusions and implications

The discrepancy between self-report regarding emotionality in the questionnaires and emotion words observed during the interviews is notable. PV increased the observational emotion report even in a sample with high emotional awareness. Future research should examine if PV can facilitate emotion reporting in a clinical sample.

---

## Introduction

Interviews are an important and commonly used tool to explore internal processes, such as emotions and cognitions [1]. There are different interview settings (e.g., face-to-face, telephone, online) and interview forms (e.g., structured interviews, focus groups, semi-structured interviews) [2,3]. Semi-structured interviews are often conducted with one person. The interviewer adheres to a predetermined set of inquiries but retains the capacity to conduct more in-depth exploration of specific themes or responses [3].

Regardless of the interview modality, various challenges may arise in the interview context. In most interviews, two strangers encounter each other. The interviewee is asked to share intimate experiences in front of an unfamiliar person. Thus, interviewees could be worried in anticipation of their first interview [4]. A particular challenge for individuals is to reveal and narrate emotions or emotional situations [5]. This is due to several factors. For example, language barriers may complicate emotional expression, societal norms may lead to emotional suppression, or an interviewee might experience difficulty in emotional introspection. [6–8].

In order to reach in-depth conversation in such interview situations, an open and trusting relationship is needed [4]. In addition, further requirements on the side of the interviewee are necessary to gain in-depth information about the individual's internal processes. First, interviewees must be able to access and observe their emotions and experiences. Second, they have to be able to find adequate words to describe these internal processes. Third, the interviewees must be willing to share these experiences with the unfamiliar person [4]. As Hiller and DiLuzio [9] put it, they must allow the interviewer to look at their "backstage". The "backstage" refers to internal processes such as emotions, ambiguities and inner conflicts. This "backstage" possibly represents the more personal and perhaps less flattering self.

Tools to help individuals put internal processes into words when being interviewed are needed to facilitate the personal report on emotions and thus to gain sincere and genuine answers [10,11]. Photovoice (PV), sometimes called *photo novella* [12], represents one way to facilitate the access to and the report of emotions. As the name suggests, PV consists of taking pictures in a first step. In a second step, people report on their emotional experiences based on the photos. The basic idea

originated from Freire [13] from the educational context and was later established mainly by Wang and Burris [14]. From its main application in community psychology [15], PV has found its way into individual settings [16] and has also been used for therapeutic interventions [17].

PV brings a number of advantages to the interview context as it is a novel way to gain insight into the inner experience of participants [5]. In particular, PV is intended to facilitate the report of emotions and support participants' self-disclosure [5,8,18]. PV creates an opportunity to express visually what is difficult to access linguistically [8] and to provide a nuanced and emotional perspective that traditional questionnaires are unable to capture [18,19]. PV is assumed to be a culturally sensitive strategy [16] and can be especially powerful for individuals who do not speak the local language [12].

The mechanism of action of PV occurs on several levels [20]. Through the physical act of taking a picture, deeper processing takes place since participants have to consider what exactly they want to photograph [11,15,16,20,21]. The physical act of photographing further allows for a contextualization of concerns, generating an understanding of how and where problems arise [19]. In addition, PV allows the person to gain distance from the experience and to reflect on it. It is thus a "reflective process of discovering and objectifying the story itself" [10]. It is further assumed that the picture allows a person to organize and emphasize what they want to share [21–23] and to access to less conscious and deeper emotions by removing natural inhibitions (e.g., shame, socially desirable answers etc.) [22].

Further mechanisms of action of PV could lie in an improved relationship between interviewer and interviewee. Indeed, PV has been found to make the interview process more interesting, creative, and collaborative [15,21,22,24]. Accordingly, it is also reported that PV leads to more self-efficacy among participants [23,25]. In the healthcare sector, self-efficacy can be defined as "the introspective conviction that one has the aptitude to take action to affect one's health" [26]. Explanations for the positive effect of PV on self-efficacy lie in the fact that PV gives participants a sense of control by bringing them into an active role, which then contributes to increased self-efficacy [18,23]. Nevertheless, studies emphasize that further research is needed on the exact relationship between PV and self-efficacy [27,28].

In summary, the potential of PV to facilitate the report on internal processes in (clinical) interviews has been qualitatively reported in many studies [5,11,29,30]. Researchers showed that people were able to express themselves more clearly using the pictures than without the pictures [20]. Even though PV is promising, it is based on too little evidence and there is a lack of controlled experimental studies investigating PV in comparison to other methods [10,27,28]. Many studies described the evaluation of PV vaguely or not at all [31]. The number of studies is limited and the quality is rather low, indicating the need for more research [27,31].

The first central aim of the current study was therefore to explore the effect of PV on the report of emotional content in an experimental design. We applied a self-report tool (assessing participants' emotional state) and an observational tool (counting the number of emotion words during an interview). We compared PV to another strategy that is typically used in cognitive-behavioral interventions to facilitate the report of emotions: an emotion journal, so-called writing group (WG), and a control group with no specific task (CG). We hypothesized that the photovoice group (PVG) would show an increased report of emotions compared to the WG and the CG. This increase was expected in both the self-report and observational tools.

Our second aim was to investigate the effect of PV on participants' perceived access of emotional content. We hypothesized that individuals in the PVG would perceive having easier access to their emotions, being able to describe their emotions more easily and reporting more self-disclosure compared to the WG and the CG.

Our third goal was to assess potential mechanisms of action related to the emotion report as secondary outcomes. We therefore investigated to what extent the interviewers' demeanor, the quality of relationship, the perceived interviewer-participant interaction, and self-efficacy were influenced by PV. We hypothesized that participants from the PVG would experience the interviewers' demeanor as more positive, the quality of relationship and the interviewer-participant interaction as better, and self-efficacy as higher compared to the WG and the CG.

## Materials and methods

### Study design and participants

This is a three-arm experimental study comparing Photovoice (PV) to an active control group (WG) and a passive control group (CG). We recruited 65 participants through flyers, university mailing lists, and on websites that promote psychological studies between November 2021 and May 2022. Participants who met the following criteria were eligible for this study: a) to be 18 years of age or older, b) to be proficient in German and, c) to self-admit worrying from time to time about some aspects of one's future. The study was approved by the local ethics committee at the Department of Psychology, Philipps-University Marburg (approval: 2021-72k) and was registered in the Open Science Framework (OSF) portal (https://osf.io/2bj63/), where the anonymized quantitative data set is also available. An optimal sample size of N = 54 for our study was determined by performing a power calculation with G*Power 3.1 (medium effect size f = 0.25, alpha = 0.05, repeated measure ANOVA with 3 groups and 2 measurements, correlation among repeated measurements r = 0.50, 1-beta = 0.90).

### Procedure

Interested participants accessed the survey directly through a web-link. The online survey was hosted on the Unipark Survey platform. Study candidates were informed about details of the study procedure, asked to provide written informed consent, and finally asked to provide some sociodemographic information and their phone number. Then, an appointment for a phone call was arranged with each participant. During the phone appointment, participants were randomly assigned to one of the three study groups using an online tool (https://zufallsgenerator.net). Participants were informed about which study arm (PVG, WG, CG) they were assigned to. They were provided with further details and instructions regarding the task that they were to complete over the following week and the interview which would take place one week after the phone call.

Participants in the PVG were asked to take pictures of future worries for one week. Participants in the WG were instructed to write down their future worries for one week. Participants in the PVG and in the WG were asked to select no more than five pictures or written worries, respectively, and to submit them before the interview would take place. In the WG, the number of keywords concerning each worry was not restricted. The CG did not receive a specific assignment and was simply notified of the interview date.

In addition to the phone call, the participants received an e-mail including information about the study procedure and the task.

After the interview, the participants were asked to fill in questionnaires online (t1). Three weeks later, participants were invited to complete an online follow-up survey (t2) (see Fig 1).

### Interview

The remote semi-structured interview took place exactly seven days after the phone call and lasted 32 minutes (*SD* = 12.94) on average. The interviews were held online through a secured medical practice software named RED medical [32]. The interviews included six open-ended questions. Participants were asked to answer each of these six questions in reference to each of the up to five chosen future worries. The questions addressed the content of the respective future worry, the meaning the worry had for the individual, emotional and physical reactions triggered by the worry, and the way individuals coped with their worry. Detailed wording of the questions can be accessed from the supplementary materials (S1 Appendix). The interview was recorded and the worry about which the participants provided the most information and spent the most time talking about was transcribed using the NVivo transcription program [33].

### Measures

**Self-reported state affect.** To measure the state affect after the interview (t1), the 20-item questionnaire Positive and Negative Affect Schedule (PANAS) [34,35] was used. Participants rated different items on how they were feeling on

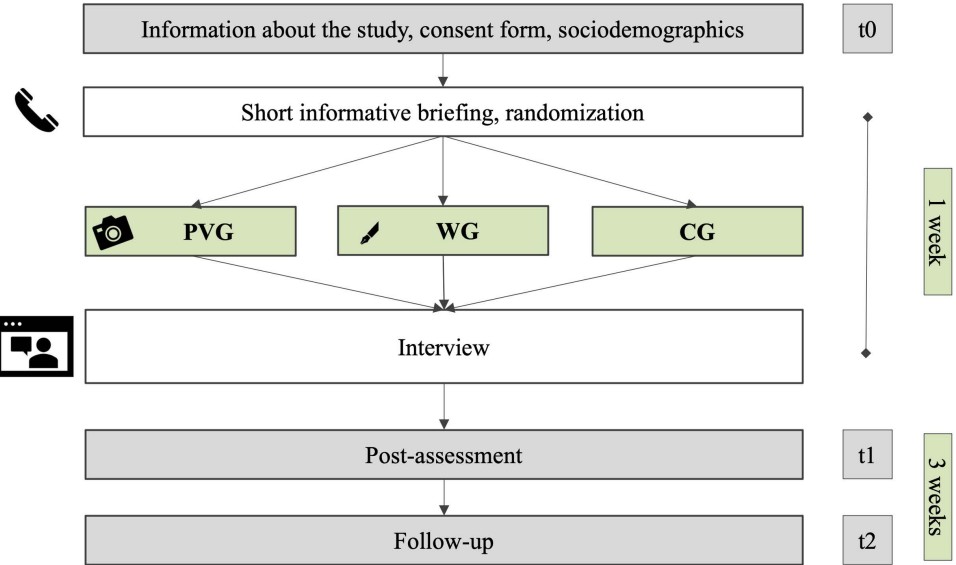

**Fig 1. Study procedure.** PVG = photovoice group, WG = writing group, CG = control group, t = time point of measurement.

a 5-point Likert scale ("*not at all*" to "*extremely*"). The 20 items describe two dimensions of affect, positive affect (e.g., interested, enthusiastic, active) and negative affect (e.g., distressed, angry, frightened). High reliability was reported in the original English-language scale (Cronbach's α was between.85 and.89 [36]). In our study, Cronbach's α was 0.83 for the positive and 0.77 for the negative affect subscale. Moreover, construct validity was demonstrated for the original scale [36] and criterion validity of the measure was evident in a German sample [37].

**Count of positive and negative emotion words during interview.** As an indicator of participants' access to and expression of emotions, the transcribed report on the future worry that participants provided the most information about was screened for emotion words. In order to analyze the interview transcripts, the 22nd version of the program *Linguistic Inquiry and Word Count* (LIWC; [38]) with the German dictionary (DE-LIWC2015; [39]) was applied. The LIWC is a computer-based text analysis program that classifies words into language categories based on a stored dictionary, counts them in terms of frequency, and finally relates them to text length [39,40]. Additionally, the LIWC indicates the percentage of positive and negative emotion words in a given text. In the context of this research project, the affective processes category with the positive (e.g., love, nice, sweet) and negative (e.g., hurt, ugly, nasty) emotions subcategories was of interest. In total, the affective processes category of the German LIWC comprises 4773 words [39].

**Perceived difficulty identifying emotions.** To measure participants' access to emotional content over the past week at the post-assessment (t1) and at follow-up (t2), the subscale of the Toronto Alexithymia Scale, "difficulty identifying emotions," which included seven items (TAS-DIE; [41,42]) was used. The items referred to the last 7 days (e.g., "*I was often confused about what emotion I was feeling*") and were rated on a 5-point scale ranging from "*strongly disagree*" to "*strongly agree.*" In our study, Cronbach's α was 0.86 at the post-assessment and 0.83 at follow-up.

**Perceived difficulty describing emotions.** To measure how individuals perceived their ability to describe their emotions over the past week at the post-assessment (t1) and at follow-up (t2), three items from the subscale "difficulty describing emotions" of the Toronto Alexithymia Scale were used (TAS-DDE; [41,42]). The participants were asked to indicate their level of agreement with statements such as, "*It was difficult for me to find the right words for my emotions.*" Statements were rated on a 5-point scale ranging from "*strongly disagree*" to "*strongly agree.*" In our study, Cronbach's α was 0.78 at the post-assessment and 0.79 at follow-up.

**Perceived fear of self-disclosure.** Participants were asked to indicate their perceived fear of self-disclosure during the interview. The subscale "fear of self-disclosure" of the German Therapeutic Alliance Scale-Revised (STA-R; [43]) was adapted for our study. For example, *therapist* was replaced by *interview partner*. Moreover, the items were rephrased in past tense to adapt the questions specifically to the interview context. Participants were asked to indicate their level of agreement with five statements (e.g., "*I found it difficult to talk openly about my thoughts and emotions with my interview partner*") using a 5-point Likert scale. The inventory was used at the post-assessment and at follow-up. In our study, Cronbach's α was 0.81 at the post-assessment and 0.82 at follow-up.

**Interviewer's demeanor.** Participants were asked to indicate how they perceived the interviewer's demeanor (INT-DEM). The first subscale of the "Measures of Rapport" by Sun et al. [44] was administered to assess this outcome. Participants indicated to what extent eleven words and phrases (e.g., friendly, easy to talk to, hard to get along with, etc.) applied to the interview partner, using a 7-point Likert scale from "*not at all*" to "*to a very great extent*". To translate the original English-language questionnaire, we applied the independent translation method [45–47]. The questionnaire was translated by several interpreters who were fluent in both English and German and who took into account the contextual meaning of the translation. The translation is included in the supplementary material (S2 Appendix). The Measure of Rapport subscale demonstrated good internal consistency in our sample (Cronbach's α = 0.86).

**Perceived interviewer-participant interaction.** Participants were asked to indicate how they perceived the interaction between themselves and the interviewer (INTERA). The second scale from "Measures of Rapport" of Sun et al. [44] was used for this purpose. Eighteen features (e.g., boring, harmonious, awkward, etc.) of the interaction were rated on a 7-point Likert scale ("*not at all*" to "*to a very great extent*"). Again, the independent translation method [45–47] was used to translate the original English-language items into German (see S2 Appendix). The subscale demonstrated good internal consistency in our sample (Cronbach's α = 0.90).

**Quality of relationship.** Four items were created by the authors to measure the quality of the relationship between the participant and the interviewer (QUAL-REL): "*I was able to engage well in the interview;*" "*During the interview I felt partly superfluous;*" "*During the interview, I actively participated in the conversation;*" and "*During the interview, I determined what I wanted to talk about.*" The items addressed to what extent the participants actively contributed to and involved themselves in the conversation. The original German-language items are provided in the supplementary material (S2 Appendix). Participants answered the items using a 5-point Likert scale from "*strongly disagree*" to "*strongly agree*". Internal consistency of these four items, based on our sample, was moderate (Cronbach's α = 0.65).

**Self-efficacy.** To measure self-efficacy, the General Self-Efficacy Scale (GSE [48]) was used. Participants indicated their agreement with 10 items on a 4-point Likert scale ("*strongly disagree*" to "*strongly agree*"). Internal consistency for the GSE ranged between Cronbach's alpha = 0.76 and 0.90 in previous studies [48] and 0.89 in our study.

**Demographics and control variables. Demographic variables, previous experience with psychotherapy, and self-reflection:** Regarding sociodemographic data, participants were asked about their age, gender, employment, level of education, and their nationality. They were asked about therapy experience and, if yes, to indicate how many hours of therapy experience they had already had. We assumed that people with therapy experience might have easier access to their emotions and be able to express them better. Moreover, two open-ended questions were used to find out how often the participants reflected on themselves and their lives. The first one focused on engaging with one's own thoughts and emotions: "*I deal a lot with myself and my thoughts and emotions.*" The second question was directed toward life and the future: "*I often engage in thoughts about life, about myself and my future.*"

**Trait anxiety.** To measure trait anxiety, we used the subscale Trait-Anxiety of the German short version of the State-Trait-Anxiety Inventory (STAI [49,50]). Participants indicated on an 8-point Likert scale how often (from "almost never" or "almost always") 10 statements occurred to them. Cronbach's α ranged between.86 and.95 in previous studies [50] and was 0.91 in our study.

**Personality.** To measure the Big Five personality dimensions in participants, the German Big Five Inventory-10 (BFI-10 [51,52]) was used. It consists of 10 items, two for each personality dimension (neuroticism, openness, conscientiousness, extraversion, and agreeableness). Participants answered the items on a five-point rating scale from "*strongly disagree*" to "*strongly agree*". The BFI-10 represents acceptable psychometric properties given its few items [53].

## Data analysis

To generate more comprehensive data, researchers in the field of PV have recommended performing method triangulation [54], that is, multiple methods are used to obtain data [55]. In our case, we contrasted the self-report measures (questionnaires) to an observational method (count of emotion words in the interview transcript). Statistical analyses were conducted using R Version 4.1.2 [56]. The significance level for all outcomes was set to $p < .05$.

All three of our hypotheses were tested by conducting ANOVAs. To compare our study groups (independent variable: PVG, WG, and CG) at t1 (post-assessment), one-way ANOVAs for each of the nine dependent variables (positive and negative affect [PANAS], word count of positive and negative emotion words [WC], perceived difficulty identifying emotions [TAS-DIE], perceived difficulty describing emotions [TAS-DDE], perceived fear of self-disclosure [STA-R], interviewer's demeanor [INT-DEM], quality of relationship [QUAL-REL], perceived interviewer-participant interaction [INTERA] and self-efficacy [GSE]) were conducted. We calculated a post-hoc contrast to compare the PVG to the WG and CG taken together (1, -0.5, -0.5). The second contrast compared the PVG to the WG only (1, -1, 0).

For non-normally distributed data, the results were analyzed with a non-parametric test (Wilcoxon test). To adjust for multiple hypothesis testing (alpha error accumulation), we applied the Bonferroni correction to our analyses.

To compare our study groups (independent variable: PVG, WG, and CG) at t2 (follow-up questionnaire), repeated measures ANOVAs were performed for each of the three dependent variables (TAS-DIE, TAS-DDE, STA-R) including the repeated measure factor time (t1: immediately after the interview and t2: 3 weeks after the interview). The main effect of study group and time and the study group*time interaction effect were analyzed.

For all outcomes, Bayesian analyses were conducted to quantify the strength of evidence regarding the null hypothesis. Bayesian analyses were used to indicate the likelihood of the null hypothesis compared to the alternative hypothesis. Bayes factors (BFs) were interpreted after Jeffreys [57]. BFs between 1 and 3 or between 0.33 and 1 indicated inconclusive or anecdotal evidence, values between 3 and 10 or between 0.10 and 0.33 indicated moderate evidence, and values greater than 10 or smaller than 0.10 indicated strong evidence. A BF of 1 indicates no evidence. Note that BFs above 1.0 support the null hypothesis, while BFs below 1.0 support the alternative hypothesis.

The BFs for the ANOVAs to analyze if there was a difference between the three groups was conducted with the anovaBF() function and default parameters (rscaleFixed = 1/2, rscaleRandom = $r$ = 1 and a sample number of 100,000). The BFs were then calculated for the sociodemographic variables, for which the contingencyTableBF() function was chosen and likewise default parameters (independent multinomial sampling type with fixed n per group).

## Results

### Participant characteristics

Fifty participants (76.9%) identified as female, 15 (23.1%) as male. The age range was between 18 and 59 ($M = 28.23$, $SD = 9.23$). The three groups (PVG, WG, CG) did not differ in any socio-demographic characteristics. The participants were highly educated, over 89.2% had completed high school or beyond, and over 70% were university students. The vast majority had German nationality (95.4%). The sample reported little preoccupation with themselves, their own thoughts, or the future (over 90% denied such concerns). This finding is surprising, considering that one inclusion criterion was to express worries about the future. One plausible explanation is that the inclusion criterion implied "worrying from time to time" whereas in the questionnaire we assessed whether participants worried "often". Please refer to Table 1 for further information.

**Table 1. Sample characteristics (N = 65).**

| Variable | (M ± SD) | | | | p value | BF01 |
|---|---|---|---|---|---|---|
| | Total N = 65 | PVG n = 22 | WG n = 21 | CG n = 22 | | |
| Age in years Range in years | 28.23 ± 9.23 18-59 | 26.81 ± 7.93 19-53 | 31.93 ± 11.92 21-59 | 26.24 ± 6.52 18-45 | 0.087 | 1.20 |
| | Number (%) | | | | | |
| Gender | | | | | 0.814 | 4.36 |
| female | 50 (76.9) | 17 (77.3) | 17 (81.0) | 16 (72.7) | | |
| male | 15 (23.1) | 5 (22.7) | 4 (19.0) | 6 (27.3) | | |
| diverse | 0 (0.0) | 0 (0.0) | 0 (0.0) | 0 (0.0) | | |
| Employment[1] | | | | | | |
| student | 47 (72.3) | 16 (72.7) | 15 (71.4) | 16 (72.7) | | |
| (self-)employed | 19 (29.3) | 6 (27.3) | 6 (28.6) | 7 (31.8) | | |
| other[2] | 3 (4.6) | 2 (9.0) | 0 (0.0) | 1 (4.5) | | |
| Level of education | | | | | 0.535 | 6.01 |
| no school diploma | 0 (0.0) | 0 (0.0) | 0 (0.0) | 0 (0.0) | | |
| secondary school | 2 (3.1) | 1 (1.5) | 1 (1.5) | 0 (0.0) | | |
| apprenticeship | 5 (7.7) | 1 (4.5) | 2 (9.5) | 2 (9.1) | | |
| high school diploma | 21 (32.3) | 9 (40.9) | 6 (28.6) | 6 (27.3) | | |
| tertiary degree | 37 (56.9) | 11 (50.0) | 12 (50.7) | 14 (63.6) | | |
| Occupation with one's thoughts and emotions[3] | | | | | 0.688 | 7.20 |
| yes or rather yes | 6 (9.2) | 2 (9.1) | 2 (9.5) | 2 (9.1) | | |
| rather no | 25 (38.5) | 11 (50.0) | 6 (28.6) | 8 (36.4) | | |
| no | 34 (52.3) | 9 (40.9) | 13 (61.9) | 12 (54.5) | | |
| Thinking often about one's own life and future[4] | | | | | 0.216 | 1.18 |
| yes or rather yes | 4 (6.2) | 2 (9.1) | 0 (0.0) | 2 (9.1) | | |
| rather no | 23 (35.4) | 11 (50.0) | 6 (28.6) | 6 (27.3) | | |
| no | 38 (58.5) | 9 (40.9) | 15 (71.4) | 14 (63.6) | | |
| Therapy experience | | | | | 0.989 | 7.19 |
| no | 38 (58.5) | 9 (40.9) | 9 (42.9) | 9 (40.9) | | |
| yes | 27 (41.5) | 13 (59.1) | 12 (57.1) | 13 (59.1) | | |
| Therapy experience(M ± SD) in sessions | 52.78 ± 57.21 | 64 ± 53.2 | 70 ± 76.4 | 24.3 ± 24.1 | | |
| Range in therapysessions | 1-250 | 7-150 | 5-250 | 1-70 | | |
| STAI (M ± SD) | 44.2 ± 15.2 | 40.8 ± 11.0 | 44.9 ± 16.8 | 46.8 ± 17.2 | 0.416 | 4.04 |
| BFI-10 (M ± SD) | | | | | | |
| Openness to Experience | 3.78 ± 0.98 | 3.68 ± 0.96 | 3.93 ± 1.16 | 3.73 ± 0.84 | 0.691 | 5.96 |
| Conscientiousness | 3.73 ± 0.85 | 3.70 ± 1.01 | 3.60 ± 0.74 | 3.89 ± 0.80 | 0.534 | 4.90 |
| Extraversion | 3.65 ± 0.90 | 3.89 ± 0.84 | 3.48 ± 0.89 | 3.59 ± 0.95 | 0.304 | 3.18 |
| Agreeableness | 3.40 ± 0.79 | 3.50 ± 0.89 | 3.40 ± 0.82 | 3.30 ± 0.68 | 0.699 | 6.00 |
| Neuroticism | 3.35 ± 1.20 | 3.20 ± 1.14 | 3.50 ± 1.15 | 3.36 ± 1.35 | 0.729 | 6.21 |

[1]Multiple answers were possible; [2]In apprenticeship, unemployed or other (e.g., retired); [3]Self constructed item: "*I deal a lot with myself and my thoughts and emotions*"; [4]Self constructed item: "*I often engage in thoughts about life, about myself and my future.*", STAI = State-Trait-Anxiety Inventory; BFI-10 = Big Five Personality Inventory-10, PVG = photovoice group, WG = writing group, CG = control group; *M* = mean, *SD* = standard-deviation, BF = Bayes factor. BF$_{01}$ demonstrates evidence strength in favor of the null hypothesis.

## Characteristics of reported worries

The future worries named by the participants were divided into categories by two independent raters. Subsequently, these were discussed and checked for inconsistencies. Two additional researchers reviewed the categorization again. A total of 13 categories were formed; the most frequent worries were welfare/illness/death of self or close living beings (mentioned 49 times) and study and profession (mentioned 44 times). Further details can be found in Table 2.

## Hypothesis 1: Self-reported state affect and count of emotion words

Contrary to our assumptions, one-way ANOVAS revealed no differences between the three study groups (PVG, WG, CG) after our intervention concerning the report of positive state affect (PANAS pos), $p = 0.872$, $BF_{01} = 7.10$ and the report of negative state affect (PANAS neg), $p = 0.533$, $BF_{01} = 4.88$, see Table 3 for details. Concerning the count of emotion words, one-way ANOVAs indicated that there were no differences in the number of positive emotion words between the three study groups, $F(2, 62) = 0.24$, $p = 0.791$, $R_{adj}^2 = -0.025$, $BF_{01} = 6.60$. However, there were differences between the study groups in negative emotions: Participants in the PVG used more negative emotion words ($M = 2.91$, $SD = 1.01$) than the WG ($M = 2.30$, $SD = 0.66$) and the CG ($M = 2.34$, $SD = 0.71$), $F(2, 62) = 3.89$, $p = 0.026$, $R_{adj}^2 = 0.08$, $BF_{01} = 0.44$. Planned contrasts revealed that the PV reported more negative emotion words compared to the two groups $t(62) = 2.79$, $p = 0.007$

**Table 2. Reported worries, ordered by frequency of mentioning.**

|    | Worry content | Number of mentioning |
|----|---------------|----------------------|
| 1  | Welfare/illness/death of self or close living beings | 49 |
| 2  | Study/profession | 44 |
| 3  | Development of relationships/ choice of partner/ future partnership | 25 |
| 4  | Anthropogenic influences on the environment and impacts thereof | 25 |
| 5  | Finances | 23 |
| 6  | FOMO (fear of missing out)/ live choices | 18 |
| 7  | War conflicts & implications | 13 |
| 8  | Loneliness | 12 |
| 9  | Social injustice/ development of society | 11 |
| 10 | Work-life balance | 9 |
| 11 | Expectations & performance | 8 |
| 12 | Desire for children & development of children | 8 |
| 13 | Other | 7 |

**Table 3. Test statistics of ANOVAs to compare the report of emotions between the PVG, the WG and the CG.**

|           | PVG M (SD) | WG M (SD) | CG M (SD) | F dfnum = 2, dfden = 62 | p | BF01 |
|-----------|-----------|-----------|-----------|-------------------------|---|------|
| PANAS pos | 2.78 (0.67) | 2.88 (0.78) | 2.85 (0.57) | 0.14 | 0.872 | 7.10 |
| PANAS neg | 1.64 (0.44) | 1.74 (0.55) | 1.82 (0.61) | 0.64 | 0.533 | 4.88 |
| WC pos    | 3.87 (1.41) | 4.04 (1.19) | 3.78 (1.19) | 0.24 | 0.791 | 6.60 |
| WC neg    | 2.91 (1.01) | 2.30 (0.66) | 2.34 (0.71) | 3.89 | **0.026** | 0.44 |

Note: PVG = photovoice group; WG = writing group; CG = control group; PANAS pos/neg = positive and negative affect schedule; WC pos = word count positive emotions, WC neg = Word count negative emotions; BF = Bayes factor. $BF_{01}$ demonstrates evidence strength in favor of the null hypothesis. P-values in bold indicate significance.

and also compared to the WG only, $t(62) = 2.48$, $p = 0.016$. As Levene's test to check the variances of the sample showed a statistical trend, $F(2, 62) = 2.99$, $p = 0.058$, the results were checked with a non-parametric test (Wilcoxon test). The Wilcoxon test confirmed the results, i.e., the PVG used more negative emotion words compared to the WG and the CG, $W = 617$, $p = 0.023$ and also compared to the WG only, $W = 310.5$, $p = 0.027$.

**Hypothesis 2: Perceived difficulty identifying and describing emotions and perceived fear of self-disclosure**

Contrary to our assumptions, one-way ANOVAS revealed no differences between the three study groups (PVG, WG, CG) after our intervention concerning the perceived difficulty identifying emotions (TAS-DIE), the perceived difficulty describing emotions (TAS-DDE), and the perceived fear of self-disclosure (STA-R), see Table 4, all $ps > 0.660$, $BF_{01} = 5.75$–7.66. The Bayes factors support the null hypothesis, indicating that it is more likely that there are no differences between the three study groups (PVG, WG, CG) than there are any differences. As can be derived from Table 4, the BFs show mostly moderate evidence for the null hypothesis. For example, the BF for the perceived difficulty identifying emotions ($BF_{01} = 6.96$) indicates that the results are almost seven times more likely to result in the null hypothesis than in the alternative hypothesis.

**Hypothesis 3: interviewer's demeanor, quality of relationship, perceived interviewer-participant interaction, and self-efficacy**

Contrary to our assumptions, one-way ANOVAS revealed no differences between the three study groups (PVG, WG, CG) after our intervention concerning the interviewer's demeanor (INT-DEM), the quality of relationship (QUAL-REL), the perceived interviewer-participant interaction (INTERA), and the self-efficacy (GSE), see Table 5, all $ps > 0.144$, $BF_{01} = 1.78$–6.67.

## Discussion

The aim of the present study was to compare photovoice (PV) with an active writing group (WG) and a passive control group (CG) in an experimental design in order to:

1) explore the effect of PV to facilitate the self-reported and observed emotion report.

2) investigate the effect of PV on participants' perceived access to emotional content.

3) assess secondary outcomes that may serve as potential mechanisms of action related to the emotion report (interviewer's demeanor, quality of relationship, perceived interviewer-participant interaction and self-efficacy).

Table 4. Test statistics of ANOVAs to compare the report of emotions between the PVG, the WG, and the CG.

| | PVG M (SD) | WG M (SD) | CG M (SD) | F dfnum = 2, dfden = 62 | p | BF01 |
|---|---|---|---|---|---|---|
| TAS-DIE | 3.77 (0.70) | 3.73 (0.76) | 3.62 (1.17) | 0.16 | 0.850 | 6.96 |
| TAS-DDE | 3.11 (0.92) | 3.10 (0.91) | 3.17 (0.96) | 0.04 | 0.965 | 7.66 |
| STA-R | 4.30 (0.59) | 4.27 (0.78) | 4.12 (0.72) | 0.42 | 0.660 | 5.75 |

Note: PVG = photovoice group; WG = writing group; CG = control group; TAS-DIE = Toronto Alexithymia Subscale "difficulty identifying emotions"; TAS-DDE = Toronto Alexithymia Subscale "difficulty describing emotions"; STA-R = Skala Therapeutische Allianz- Revised; BF = Bayes's factor. $BF_{01}$ demonstrates evidence strength in favor of the null hypothesis.

A repeated measures ANOVA was performed on study group (PVG, WG, and CG) and time (t1: immediately after the interview and t2: 3 weeks after the interview) concerning perceived difficulty identifying and describing emotions and perceived fear of self-disclosure. The only main effect of time was shown regarding the extent of self-disclosure, $F(1, 55) = 38.11$, $p < 0.001$, $\eta_g^2 = 0.16$. The participants showed more fear of self-disclosure at the post-assessment ($M = 4.31$, $SD = 0.62$) than at the follow-up ($M = 3.68$, $SD = 0.87$), independent of the study group. All other variables were not significant and there were no interaction effects, $p > 0.05$.

**Table 5. Test statistics of ANOVAs to compare the secondary outcomes between the PVG, the WG and the CG.**

| | PVG M (SD) | WG M (SD) | CG M (SD) | F dfnum = 2, dfden = 62 | p | BF01 |
|---|---|---|---|---|---|---|
| INT-DEM | 5.81 (0.60) | 6.18 (0.56) | 5.79 (0.91) | 2.00 | 0.144 | 1.78 |
| QUAL-REL | 4.38 (0.62) | 4.62 (0.36) | 4.40 (0.65) | 1.23 | 0.301 | 3.16 |
| INTERA | 4.38 (0.62) | 4.62 (0.36) | 4.40 (0.65) | 1.59 | 0.212 | 2.41 |
| GSE | 27.6 (4.02) | 26.6 (6.90) | 27.6 (6.48) | 0.22 | 0.803 | 6.67 |

Note: PVG = photovoice group; WG = writing group; CG = control group; INT-DEM = Interviewer's demeanor; QUAL-REL = quality of relationship; IN-TERA = perceived interviewer-participant interaction; GSE = general self-efficacy scale; BF = Bayes's factor. $BF_{01}$ demonstrates evidence strength in favor of the null hypothesis.

Concerning the effect of PV to facilitate emotion report, the self-report measurements showed no differences between the PVG, the WG and the CG. The interview transcripts, however, indicated that the PVG used more negative emotion words than the WG and the CG.

One potential reason for this discrepancy is that the perception and presence of emotions diverged. As a result, a person might feel that they were not experiencing certain emotions, whereas it became noticeable in the interview content. Previous research has already noted this discrepancy between self-report and observational measures. For example, Güney et al. [58] reported that although individuals reported suppressing their emotions, the negative emotions showed up physically through muscle movements, vocal and postural behavior. It is possible that while access to more emotions was not reported in the questionnaires, the difference was made visible at the linguistic level (i.e., in the transcribed interview). This means that PV may implicitly lead to people being more likely to talk about negative emotions, but without consciously noticing it and thus without being able to express it in a conventional survey. This result underscores the importance of collecting and reconciling data at multiple modes (for example, collect and match questionnaires and interview transcripts or questionnaires and body responses simultaneously).

One could critically interject that the PVG may have exhibited a more negative mood at the beginning of the interview and consequently reported more negative emotion words. However, this is contradicted by the fact that the three groups did not show any differences in the number of the following characteristics: anxiousness, mood immediately after the interview, sociodemographic variables, and personality traits. Thus, it can be cautiously assumed that PV facilitated the reporting of negative emotional experiences even though the effect was small.

We further investigated the effect of PV on participants' perceived access of emotional content. Contrary to our hypotheses, there were no differences between the three groups. Identifying and expressing emotions is an extremely complex process that is influenced by various factors such as personality, cultural influences, and semantic knowledge about one's own emotions [59]. Even if the emotion report was facilitated on an observational level, additional steps are necessary for its reporting. First, participants must be able to perceive their increased access, and second, they must be able to report it back. Also, self-reports regarding access to feelings are subject to various influences, such as one's self-esteem, social desirability bias, etc. [59]. In addition, the questions addressed the week in which PV was applied. Future studies should also explore whether emotional content access was specifically facilitated during the interview.

Finally, we assessed secondary outcomes related to the emotion report (interviewer's demeanor, quality of relationship, perceived interviewer-participant interaction, and self-efficacy). Contrary to our hypotheses, none of the self-report measures on these secondary outcomes showed any differences between the three groups.

If only the answers in the self-report measures were considered, it could be argued that PV did not represent a benefit over other methods. Further, the Bayesian analyses supported to a moderate extent that there were likely no differences between the PVG, the WG, and the CG. This is consistent with a recent meta-analysis [27] that showed that PV does not

result in observable mental health outcomes or lead to greater self-efficacy. However, it is in contrast to many PV studies [8,16,18], which qualitatively attribute several advantages to PV (e.g., better access to emotions, more self-efficacy, etc.).

Explanations for the lack of group differences relate, among other factors, to the methodological implementation of the PV study. First, unlike most other studies, we used an experimental design and have set the time frame at one week. Previous studies typically asked participants to take pictures over a longer period of time (median of 3 months [15]) and subsequently discuss them in groups. It may be that one week was not enough to reach the necessary intensity and thus the resulting benefits. Second, we did not assess time and commitment (how much time was spent daily on picture taking and on how many days). Third, the task (taking pictures on future worries) was defined very vaguely in order not to limit the creativity of the participants. It is possible that a clearer and more specific task would have facilitated more in-depth engagement. A meta-analysis on *Expressive Writing* showed that the effects of a writing intervention were greater when the topic was more specific [60]. Thus, for future studies, it might be useful to formulate the picture-taking task to be more directive and to ask participants about the time actually spent on picture taking. Fourth, all three groups were asked the same questions during the interview in order to maintain comparability between the groups. This had the disadvantage that the picture was not explicitly used during the interview. It is likely that it would have been necessary to bring the pictures more actively into the interviews to use their full potential. In previous studies, most researchers used the SHOWeD method as guidance for the interviews [5,12,61]. This method starts the conversation with the question: *What do you see here?* We purposely opted not to use the SHOWed method in order to standardize the procedure over all groups. Using this method could have improved the expected effect of PV, i.e., it would have facilitated the access and expression of emotions through the explicit use of the pictures.

Fifth, another explanation for the lack of group differences in the self-report measures may be related to sample characteristics. Although the inclusion criteria were defined broadly, the sample was relatively homogeneous. The majority was highly educated, young, and mostly identified as female. Studies show that these characteristics are associated with increased emotional awareness [62]. Hence, it can be surmised that most participants did not need the support of the picture to report their future worries.

The perspectives and experiences of highly educated participants likely differ significantly from those of other social groups, which may also be reflected in the language and emotions they express. The lack of representativeness or our sample regarding educational level and gender limits the generalizability of our results. This highlights the need to replicate the study in other samples to capture a wider spectrum of experiences and emotions.

However, it is important to highlight that despite the sample's high homogeneity, we observed a discrepancy between self- and observational report.

Moreover, the majority (over 90%) said they were not concerned about the future or their lives in general. Thus, the question arises whether a group that would be more distressed might benefit more from the intervention. Therefore, PV may be more appropriate and beneficial in a clinical sample. Studies also suggest that PV could be particularly beneficial in samples that have limited access to the dominant language [27].

In addition to these explanatory approaches, some general limitations of the study are highlighted here. First, the number of participants was relatively small, which may have resulted in small effects not being detected. Second, PV may not be a "one size fits all" method, but could be appropriate for certain individuals. Researchers have argued that PV is only appropriate for people who can express themselves through photographs and who are interested in this method [20]. Further research is needed to make statements about relationships between benefits from PV and participant characteristics.

Despite these limitations, several strengths of the study should be highlighted. To the best of our knowledge, it is the first study comparing PV in an experimental design with another active control group and a passive group. Further, using different methods (interview transcripts, self-report in questionnaires) enabled a more holistic overview and an increase in reliability and validity [55]. It can be emphasized that we investigated various different outcomes (relationship quality, self-efficacy, and others) that were associated with PV in previous research. These factors have often been mentioned

in qualitative studies, but have rarely been investigated quantitatively to date. Furthermore, the innovative study design holds high external validity. Although the effort for participants was considerable (several contacts with the investigators, task over one week, follow-up survey), a variety of people could be recruited for the study. It was shown that the task (to take a picture) can be implemented well in an online individual setting and via video conference. Further, the feedback on the study from participants should be highlighted. Participants emphasized that, "PV could help to find new ways to deal with worries." Finally, the word count on emotion words indicates that the participants in the PVG used more emotionally negative words compared to the WG and the CG. This could be an indication that PV facilitates the experiential reporting of more emotionally difficult topics.

## Conclusions and implications

In conclusion, the present study lays out a well-established framework for how PV can be investigated. We demonstrated how PV can be implemented in an online and individual setting and how it can be used in interviews addressing emotionally challenging topics.

We point out important limitations that can be considered and adapted in future research. We argued that PV could be particularly suitable for samples of individuals who have difficulties in developing awareness of as well as reporting their emotions. We assume that extending the time frame for PV-related tasks such as taking the pictures and journaling experiences with the PV method would be likely to increase its effects. We also think that for future research and the application of PV, providing participants with clearer instructions on the task would improve the effects of PV. We did not find any differences in any of the self-report measures such as the perceived access of emotional content, nevertheless, the study demonstrates that PV increased the observed emotion report even in a sample of individuals who have a presumably high emotional awareness. Future research should examine whether PV can facilitate the observational and self-reported emotion report in a clinical sample, especially in individuals who tend to experience difficulties in accessing their emotions. If PV facilitates the report of emotions, this would have important practical implications. In an interview or clinical setting, PV could be used to gather sensitive information and make it easier for people to talk about their internal processes. Especially in disorders where access to emotions is more difficult (e.g., in somatoform disorders [58]), PV could serve as a bridge to communicate the inner experience more easily. The study provides preliminary evidence that PV facilitates the report of emotions, but more research is needed to determine to what extent and under what circumstances PV facilitates reporting on challenging issues.

## Supporting information

**S1 Appendix. Interview guide.**
(PDF)

**S2 Appendix. Questionnaires.**
(PDF)

## Acknowledgments

We thank Sarah Daehler for proofreading our manuscript. We would further like to thank Niclas Damm for his support in data collection and study design. We are also grateful to all the participants who shared their experiences with us.

## Author contributions

**Conceptualization:** Selina Studer, Maria Kleinstäuber, Cornelia Weise.

**Data curation:** Selina Studer.

**Formal analysis:** Selina Studer.

**Methodology:** Selina Studer, Maria Kleinstäuber, Cornelia Weise.

**Project administration:** Selina Studer.

**Supervision:** Maria Kleinstäuber, Cornelia Weise.

**Writing – original draft:** Selina Studer.

**Writing – review & editing:** Selina Studer, Maria Kleinstäuber, Cornelia Weise.

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
