## [Decision Letter · Decision Letter 0]

2 Sep 2024

PONE-D-23-40707Using Photovoice to facilitate the report of emotions in an interview setting. 

An experimental study.PLOS ONE

Dear Dr. Studer,

Thank you for submitting your manuscript to PLOS ONE. After careful consideration, we feel that it has merit but does not fully meet PLOS ONE’s publication criteria as it currently stands. Therefore, we invite you to submit a revised version of the manuscript that addresses the points raised during the review process.

We look forward to receiving your revised manuscript.

Kind regards,

Cho Lee Wong, PhD

Academic Editor

PLOS ONE

Journal requirements: 1. When submitting your revision, we need you to address these additional requirements. Please ensure that your manuscript meets PLOS ONE's style requirements, including those for file naming. The PLOS ONE style templates can be found at https://journals.plos.org/plosone/s/file?id=wjVg/PLOSOne_formatting_sample_main_body.pdf and https://journals.plos.org/plosone/s/file?id=ba62/PLOSOne_formatting_sample_title_authors_affiliations.pdf. 2. We note that you have indicated that there are restrictions to data sharing for this study. PLOS only allows data to be available upon request if there are legal or ethical restrictions on sharing data publicly. For more information on unacceptable data access restrictions, please see http://journals.plos.org/plosone/s/data-availability#loc-unacceptable-data-access-restrictions.  Before we proceed with your manuscript, please address the following prompts: a) If there are ethical or legal restrictions on sharing a de-identified data set, please explain them in detail (e.g., data contain potentially identifying or sensitive patient information, data are owned by a third-party organization, etc.) and who has imposed them (e.g., a Research Ethics Committee or Institutional Review Board, etc.). Please also provide contact information for a data access committee, ethics committee, or other institutional body to which data requests may be sent. b) If there are no restrictions, please upload the minimal anonymized data set necessary to replicate your study findings to a stable, public repository and provide us with the relevant URLs, DOIs, or accession numbers. For a list of recommended repositories, please seehttps://journals.plos.org/plosone/s/recommended-repositories. You also have the option of uploading the data as Supporting Information files, but we would recommend depositing data directly to a data repository if possible. We will update your Data Availability statement on your behalf to reflect the information you provide. 3. In the online submission form, you indicated that [Data cannot be shared publicly because of sensitive content (e.g. worries reported in the interview contain personal and potentially identifying information).Numeric answers to the questionnaire are available on request from the author.]. All PLOS journals now require all data underlying the findings described in their manuscript to be freely available to other researchers, either 1. In a public repository, 2. Within the manuscript itself, or 3. Uploaded as supplementary information.This policy applies to all data except where public deposition would breach compliance with the protocol approved by your research ethics board. If your data cannot be made publicly available for ethical or legal reasons (e.g., public availability would compromise patient privacy), please explain your reasons on resubmission and your exemption request will be escalated for approval. 

Additional Editor Comments:

This study aimed to investigate the effectiveness of PV to aid in emotional processing in comparison to other methods. The topic is interesting and innovative. Below are a few comments for consideration.

Methods

1. Please elaborate on the sample size calculation.

Results

1. Given the different age ranges and employment status, but small sample sizes, would this have an impact on the results?

Reviewers' comments:

Reviewer's Responses to Questions

**Comments to the Author**

1. Is the manuscript technically sound, and do the data support the conclusions?

Reviewer #1: Yes

Reviewer #2: Yes

2. Has the statistical analysis been performed appropriately and rigorously?

Reviewer #1: Yes

Reviewer #2: Yes

3. Have the authors made all data underlying the findings in their manuscript fully available?

Reviewer #1: No

Reviewer #2: Yes

4. Is the manuscript presented in an intelligible fashion and written in standard English?

Reviewer #1: Yes

Reviewer #2: Yes

5. Review Comments to the Author

Reviewer #1: Thank you for having me review this manuscript. The work is in the important field of psychology and the manuscript is well written. This complements the growing knowledge about the photovoice approach. A few comments to improve the manuscript here below:

Abstract.

1. Please add information about gender and mean age. For mean age, provide its standard deviation.

Discussion.

1. Please add further clarification to the limitations section. Your sample included highly educated participants, most of whom were students. Moreover, you had fewer male participants, which could imply an insufficient heterogeneity of the sample.

Conclusion.

1. From line 503, page 23 to line 515, page 23 you are talking about implications. Please add the implications section.

2. In the conclusion section, please describe your findings more specifically.

Reviewer #2: Its a good topic of research. Following are some observations:

Don't use full stop in the title.

Were universally same emotions were checked?

Validity of BFI was already established?

Which software has been used to analyze data?

6. PLOS authors have the option to publish the peer review history of their article (what does this mean? ). If published, this will include your full peer review and any attached files.

**Do you want your identity to be public for this peer review?** For information about this choice, including consent withdrawal, please see our Privacy Policy .

Reviewer #1: **Yes: ** Viktoryia Karchynskaya

Reviewer #2: No

---

## [Author Response · Author response to Decision Letter 1]

10 Nov 2024

PEER REVIEW REPLY

ACADEMIC EDITOR

Authors’ reply:

Thank you for your feedback on PLOS ONE's style requirements. We have formatted our manuscript according to the PLOS ONE’s style requirements.

2. We note that you have indicated that there are restrictions to data sharing for this study. PLOS only allows data to be available upon request if there are legal or ethical restrictions on sharing data publicly.

3. In the online submission form, you indicated that [Data cannot be shared publicly because of sensitive content (e.g. worries reported in the interview contain personal and potentially identifying information).

Numeric answers to the questionnaire are available on request from the author.].

Authors’ reply:

Thank you for your important feedback on data sharing.

The interview transcripts include sensitive and identifiable personal information (e.g., individuals’ worries), which we unfortunately cannot share. However, we uploaded our anonymized quantitative data set to OSF where they will be accessible to other researchers.

To prevent the risk of re-identifying participants due to the small sample size and specific nature of the questions, we followed the ethics committee's recommendation and removed certain socio-demographic data from the uploaded dataset.

Authors’ reply:

Thank you for your remark. We have rechecked the reference list and confirm that our reference list is complete and correct; no changes have been necessary.

This study aimed to investigate the effectiveness of PV to aid in emotional processing in comparison to other methods. The topic is interesting and innovative. Below are a few comments for consideration.

Authors’ reply:

Thank you very much for your appreciative and very constructive feedback!

Methods

1. Please elaborate on the sample size calculation.

Authors’ reply:

We have to apologize, we missed to report our power analysis in our original submission. We added the following paragraph on our power calculation to the Methods section (Study design and participants):

The optimal sample size of N=54 was determined by performing a power calculation with G*Power 3.1 (medium effect size f = 0.25, alpha = 0.05, repeated measure ANOVA with 3 groups and 2 measurements, correlation among repeated measurements r = 0.5, 1-beta = 0.90).

Results

1. Given the different age ranges and employment status, but small sample sizes, would this have an impact on the results?

Authors’ reply:

Thank you for your insightful question. Our hypotheses and analyses focus on the comparison of our study groups. Given that we randomized our participants and baseline analyses confirm that our randomization was successful (no between-group differences regarding socio-demographic variables were found), the range of age of age does not have impact on the interpretation of our findings. In the contrary, the broad age range in each of our study arms increases the representativeness of our sample.

The increased number of students in our sample has impact on the generalizability of our findings. This problem has been discussed as limitation of our trial in the Discussion section (see p. 21). It does not have impact on the interpretation of our analyses of between-group effects, because here again, randomization was successful and there are no differences in employment status between our study arms.

REVIEWER #1

Reviewer #1: Thank you for having me review this manuscript. The work is in the important field of psychology and the manuscript is well written. This complements the growing knowledge about the photovoice approach. A few comments to improve the manuscript here below:

Authors’ reply:

Thank you very much for your appreciative and very constructive feedback!

Abstract.

1. Please add information about gender and mean age. For mean age, provide its standard deviation.

Authors’ reply:

Thank you for your feedback. We have updated the manuscript as suggested and included the information on gender and mean age to the abstract.

Discussion.

1. Please add further clarification to the limitations section. Your sample included highly educated participants, most of whom were students. Moreover, you had fewer male participants, which could imply an insufficient heterogeneity of the sample.

Authors’ reply:

Thank you for the valuable feedback. We have revised the relevant section and further emphasized the limitations of the representativeness of our sample (see p. 21):

The perspectives and experiences of highly educated participants likely differ significantly from those of other social groups, which may also be reflected in the language and emotions they express. The lack of representativeness or our sample regarding educational level and gender limits the generalizability of our results. This highlights the need to replicate the study in other samples to capture a wider spectrum of experiences and emotions.

Conclusion.

1. From line 503, page 23 to line 515, page 23 you are talking about implications. Please add the implications section.

Authors’ reply:

Thank you for pointing this out. You are absolutely right that this section addresses the implications. We have revised the title to clarify that the implications are discussed in this place.

2. In the conclusion section, please describe your findings more specifically.

Authors’ reply:

Thank you for your feedback. We have revised the Conclusion section to present the findings in a more specific way. The updated paragraph encompassing our conclusions can be found on page 23. We hope this provides a more specific summary of the results.

REVIEWER #2

Reviewer #2: Its a good topic of research. Following are some observations:

Authors’ reply:

Thank you very much for your appreciative and very helpful feedback!

Don't use full stop in the title.

Authors’ reply:

We have revised the title according to your suggestions and removed the full stop. The title reads now as: Using Photovoice to facilitate the report of emotions in an interview setting: An experimental study

Were universally same emotions were checked?

Authors’ reply:

Thank you for this interesting comment. We did not explicitly examine universal emotions. The software we used (LIWC) identifies emotion words and counts the frequency of positive and negative emotion words, but it does not distinguish between universal and culturally specific emotions. It would be an interesting research question to explore whether photovoice has a specific impact on universal emotions. However, our sample was very homogeneous regarding cultural background, meaning that our data set is probably not appropriate to examine specific effects on universal emotions. Our study focuses on whether the report of any type of emotions was facilitated. In future research, with a more culturally diverse sample, we could investigate differential effects of Photovoice on the report of universal and culturally specific emotions.

Validity of BFI was already established?

Authors’ reply:

Thank you for this important comment. The article published by Rammstedt and John (2007) specifies the validity of the BFI . The researchers concluded that the “results indicate that the BFI-10 scales retain significant levels of reliability and validity”.

Due to space constraints and the need for consistency, we have not listed the validity of each questionnaire individually. However, readers of our manuscript are referred to the article by Rammstedt and John in the reference list of our manuscript.

Which software has been used to analyze data?

Authors’ reply:

Thank you for your question to clarify how we analyzed our data. Statistical analyses were conducted using the software R Version 4.1.2 The reference is: R Core Team. R: A language and environment for statistical computing. Vienna, Austria: R Foundation for Statistical Computing; 2022. We added this information to our manuscript on page 12.

---

## [Decision Letter · Decision Letter 1]

16 Mar 2025

Using Photovoice to facilitate the report of emotions in an interview setting: An experimental study

PONE-D-23-40707R1

Dear Dr. Studer,

We’re pleased to inform you that your manuscript has been judged scientifically suitable for publication and will be formally accepted for publication once it meets all outstanding technical requirements.

Kind regards,

Cho Lee Wong, PhD

Academic Editor

PLOS ONE

Additional Editor Comments (optional):

Reviewers' comments:

Reviewer's Responses to Questions

**Comments to the Author**

1. If the authors have adequately addressed your comments raised in a previous round of review and you feel that this manuscript is now acceptable for publication, you may indicate that here to bypass the “Comments to the Author” section, enter your conflict of interest statement in the “Confidential to Editor” section, and submit your "Accept" recommendation.

Reviewer #3: All comments have been addressed

Reviewer #4: All comments have been addressed

2. Is the manuscript technically sound, and do the data support the conclusions?

Reviewer #3: Yes

Reviewer #4: Yes

3. Has the statistical analysis been performed appropriately and rigorously?

Reviewer #3: Yes

Reviewer #4: Yes

4. Have the authors made all data underlying the findings in their manuscript fully available?

Reviewer #3: Yes

Reviewer #4: Yes

5. Is the manuscript presented in an intelligible fashion and written in standard English?

Reviewer #3: Yes

Reviewer #4: Yes

6. Review Comments to the Author

Reviewer #3: (No Response)

Reviewer #4: All earlier recommendations were revised, making the manuscript more clear, especially the methods and limitations.

7. PLOS authors have the option to publish the peer review history of their article (what does this mean? ). If published, this will include your full peer review and any attached files.

**Do you want your identity to be public for this peer review?** For information about this choice, including consent withdrawal, please see our Privacy Policy .

Reviewer #3: No

Reviewer #4: No

---

## [Editor Report · Acceptance letter]

PONE-D-23-40707R1

PLOS ONE

Dear Dr. Weise,

I'm pleased to inform you that your manuscript has been deemed suitable for publication in PLOS ONE. Congratulations! Your manuscript is now being handed over to our production team.

Kind regards,

on behalf of

Dr. PLOS Manuscript Reassignment

Staff Editor

PLOS ONE